# High Input of Nitrogen Fertilization and Short Irrigation Frequencies Forcefully Promote the Development of Verticillium Wilt of Olive

**DOI:** 10.3390/plants11243551

**Published:** 2022-12-16

**Authors:** Mario Pérez-Rodríguez, Antonio Santos-Rufo, Francisco Javier López-Escudero

**Affiliations:** 1Excellence Unit ‘María de Maeztu’ 2020-23, Department of Agronomy, Campus de Rabanales, University of Cordoba, 14071 Cordoba, Spain; 2Department of Agroforestry Sciences, ETSI University of Huelva, 21007 Huelva, Spain

**Keywords:** fertigation, infested soil, *Olea europaea*, *Verticillium dahliae*, watering

## Abstract

It is known that high N doses, N/K imbalances, and frequent irrigation favor Verticillium wilt. The influence of fertilization and its interaction with the frequency of irrigation on the development of Verticillium wilt of olive (VWO) has been evaluated. A split-split-plot design in microplots with two naturally infested soils of different texture was established for studying three fertilization treatments (NO_3_Ca, NPK and without fertilization), plus two irrigation frequencies (daily and deficit). The treatments were applied by means of fertigation, evaluating the susceptible cultivar Picual. Final disease incidence in plants subjected to NO_3_Ca daily treatment was 100% regardless of soil texture. However, final mortality in these plants was 37% and 85.2% in clay and sandy loam soils, respectively. In addition, the area under the disease progress curve values were significantly higher (49.1%) in plants subjected to NO_3_Ca fertilization compared to those not fertilized or fertilized with N-P-K when plants were grown in clay soil. This value in the sandy loam soil was significantly higher in the NO_3_Ca daily irrigation treatment (94.3%), followed by the N-P-K-daily treatment (61.1%) which also was significantly higher than the unfertilized daily, N-deficit and NPK-deficit treatments (37.8, 42.6 and 44.9%, respectively). The plants submitted to unfertilized-deficit treatment reached the lowest value (9.6%). In this work it can be concluded that the application of fertilizer or the application of fertilizer with daily irrigation in naturally infested soils increases the development of VWO in Picual.

## 1. Introduction

For the last four decades, Verticillium wilt of olive (VWO), caused by the soil-borne pathogen *Verticillium dahliae* Kleb., has become the most threatening disease of olive orchards, killing a huge number of productive trees in the main olive oil producing countries [1,2,3,4,5]. An integrated control strategy is strongly recommended since the application of measures individually is not effective. Integrated disease management comprises preventive measures applied before planting (use of pathogen-free plants and plantation on non-infested soils), combined with measures after planting that are mainly aimed to prevent the pathogen from arriving at the olive orchards or to reduce its spread and efficacy in causing infections. In this regard, it is essential to clarify the individual influence of the main culture practices carried out in modern olive orchards on disease development and interactions with each other. Two of the most important of these cultural practices are fertilization and irrigation, particularly in the modern intensive plantation methods that use fertigation systems [6].

The influence of fertilization in vascular disease has been studied in a wide range of crops [7,8,9,10]. However, most studies have been conducted in herbaceous crops. Several authors have reported that low nitrogen (N) levels reduce susceptibility to Verticillium wilts in lucerne [11], hop [12,13] and eggplant [14]. In contrast, high nitrogen content combined with potassium (K) deficiency may cause an increase in disease severity [9]. For instance, Presley and Dick (1951) [15] showed that when K is deficient, application of phosphate and low N level increased wilt in cotton. Nevertheless, these interactions can be complex. For example, according to studies of Young et al. (1959) [16], superphosphate application alone had no effect on wilt in cotton fields.

In woody species there is little available information. However, in the same way as for herbaceous plants, it has been reported that increasing nitrogen fertilization promotes Verticillium wilt of maple [17] and that incidence of Verticillium wilt in pistachio is significantly associated with deficiencies of potassium in soils with low inoculum densities of *V. dahliae* [18]. In contrast, no effects have been seen in cacao seedlings under altered levels of phosphorous [19]. In olive, it is assumed by farmer and agronomists, that onset and development of the disease is exacerbated by excessive nitrogen fertilization. In this crop tree, López-Moral et al. (2022) [20] recently posited that the effect of excess nitrogen (N) or imbalances in N-potassium (K) favor VWO in rooted olive cuttings cultivated in pots with an artificially pathogen-colonized substrate. In this research, treatments with N and Na (sodium) excess, and with N and Na excess and K deficiency, decreased the germination of the fungus microsclerotia and the disease progress. Nevertheless, there is no research published on this matter under field conditions, which provides scientific evidence to explain how this agricultural practice affects disease prevalence and severity in olive. 

Similarly, the beneficial effect of limiting the amount of water applied to the field to reduce Verticillium wilt epidemics has been widely studied in herbaceous hosts of the pathogen, such as potato [21], cauliflower [22], eggplant [23], and cotton [24]. However, some attempts have been made on woody hosts to prove the detrimental effect of excess water on Verticillium wilt of cacao [19], apricot [25] and maple 17]. In olive, several recent reports of long-term field experiments and field surveys in growing areas affected by VWO have clarified the influence of watering on the disease under different experimental conditions [26,27,28,29]. When experiments were conducted in infested microplots and in naturally infested fields by the pathogen, daily irrigation schedule strongly encouraged the onset and development of the disease in the susceptible cultivar Picual when compared to the rest of treatments (weekly, biweekly and deficit) or the use of other olive cultivar with higher resistance level as ‘Arbequina’ and ‘Frantoio’. These results led the authors to recommend the extension of irrigation frequencies in susceptible cultivars to reduce the incidence of disease. Additionally, surveys conducted during 2012 and 2013 comprising 70 olive orchards in a representative olive growing area in the Guadalquivir Valley under disease risk conditions showed that 47.1% of olive plantations classified as high irrigated (mean water consumption of 1900 m^3^/ha), showed the highest disease incidence (21.42%). Plantations watered with lesser doses (the 28.6%) (mean water consumption of 725 m^3^/ha) showed a disease incidence significantly lower, accounting for 13.3%. Finally, dry-land olive orchards (24.3%) showed the lowest Verticillium wilt incidence (4.0%). Therefore, these data show the reduction in watering dosage could be an advisable cultural practice for controlling the disease in affected olive orchards [28]. Moreover, in 70% of olive visited farms, disease symptoms onset coincided with the conversion of the orchards from dry-land into irrigation, and disease incidence was influenced by the number of years of irrigation, following a linear trend [28].

The objective of this research work was to evaluate de influence of the interaction of different fertilization formulations and different irrigation frequencies on the onset and development of Verticillium wilt of olive in the main olive cultivar Picual.

## 2. Results

### 2.1. Disease Symptoms and Pathogen Isolation from Affected Tissues

During the experiment, the affected plants exhibit apoplexy and slow decline [1] (Appendix A). During the fall, olive plants showed chlorosis, dieback, rolling of the leaves without defoliation of green leaves, plants completely wilted and plants with heavy defoliation of branches. The pathogen was recovered from most of the shoots which showed symptoms. All isolates recovered from symptomatic plants in both soils were identified as the defoliating pathotype that correspond to highly virulent isolates. 

### 2.2. Inoculum Density of V. dahliae in the Soil and Disease Progression

Soil analyses showed that the pathogen remained viable in both soils throughout for 11 months. The inoculum density in the containers with clay soil and with sandy loam soil was homogeneous, with average values of 12.0 and 4.5 microsclerotia per gram of air-dried soil, respectively. No significant differences (*p* = 0.87) were detected in the initial and final amounts of soil pathogens between different treatments in both soils. Thus, the ID went from 0.0 to 40.0 microsclerotia per gram of air-dried soil. 

The combined analysis (Split-split-plot) of variance for final values of RAUDPC and Severity showed that the four analyzed factors (Soil, fertilizer, soil-fertilizer interaction, and irrigation) were significant, with *p* values < 0.05 respectively. Significance of the interaction was due to use two different soils. For that, each soil was studied separately. 

The onset of symptoms development in clay soil was recorded 31 weeks after planting in microplots unfertilized when plants were irrigated daily. However, in sandy-loam soil the first symptoms were also observed 31 weeks after planting, but in daily irrigated microplots unfertilized and fertilized with nitrogen (Table 1 and Table 2; Figure 1). 

In sandy-loam soil, the appearance of first symptoms was significantly earlier in plants fertilized (average values of 37.2 and 36.4 weeks in NPK and N treatments, respectively) compared with plants unfertilized (average value of 40.2 weeks). However, differences were not detected (from 41.7 to 39.4 weeks) in clay soil (*p* > 0.05; Table 1 and Table 2).

During fall 2015, in clay soil, Disease Incidence (DI) increased faster in plants under N-daily treatment reaching a value of 77.7% in 8 weeks (from 15 September to 1 November) (Figure 1). At this time, the DI in other treatments did not reach values greater than 37%. At the end of the experiment, the analysis of variance for DI showed that only irrigation factor was significant (*p* = 0.0117; Data not shown): 80.2% and 50.6% of mean incidence in plants daily irrigated and with deficit irrigation frequency, respectively. In sandy loam-soil, a serious epidemic developed in plants under N-daily treatment (Figure 1). At firsts of November DI was 100%. On that date, the treatments NPK-daily, NPK-deficit and N-deficit reached values of DI higher than 80%. However, the disease was considerably delayed in plants unfertilized with a deficit irrigation. The final value of DI in this treatment was 55.6% (Table 2), significantly lower than the rest of treatments (*p* = 0.0494). At the end of the assessment period, significant differences were found among irrigation treatments (*p* = 0.0350; Data not shown): 98.8% of incidence in plants daily irrigated and 82.7% in plants with deficit irrigation frequency.

After 11 months of observations in clay soil, the percentage of trees killed by the pathogen was significantly (*p* = 0.0256) lower in plants unfertilized and fertilized with NPK (average Mortality value of 12.9% and 11.1%, respectively) than plants fertilized with N (average value of 31.4%) (Table 1). However, in sandy-loam soil this percentage was significantly higher (*p* = 0.0134) in plants fertilized with NPK and N (average value of 48.1 and 53.7, respectively) than plants unfertilized (average value of 12.9%) (Table 2). In this soil, the mortality in ‘Picual’ plants that were irrigated daily reached values significantly higher (*p* = 0.0205) compared with ‘Picual’ plants under deficit irrigation (53.08% and 23.45%, respectively). In clay soil, no significant differences were detected among irrigation treatments (*p* = 0.1296).

At the end of the assessment period, in sandy loam soil, the mean time from symptom onset to plant death (MM parameter) was significantly shorter in trees under the treatment N-daily (3.7 weeks) than for plants of the other treatments (Table 2). In clay soil no differences were found (Table 1). 

Significant differences regarding the final value of RAUDPC among treatments varied depending on the soil. In clay soil, the factors fertilizer and irrigation were significant, with *p* value = 0.0119 and *p* value = 0.0351, respectively. The mean value of RAUDPC calculated in plants N-fertilized were significantly higher (49.1%) than the mean value obtained in plants unfertilized, and plants fertilized with NPK. In sandy-loam soil, the three analyzed factors (fertilizer, irrigation and interaction fertilizer-irrigation) were significant (*p* < 0.05). Picual trees that were daily irrigated and fertilized with N reached RAUDPC values of 94.3% (58.4% significantly higher than the mean value obtained in the rest of treatments) followed by plants daily irrigated and fertilized with NPK, which reached RAUDPC values of 61.1%. However, unfertilized-daily, N-deficit and NPK-deficit treatments reached values of 37.8, 42.6 and 44.9%, respectively. The plants submitted to unfertilized-deficit treatment reached the lowest value, with a RAUDPC of 9.6% (Table 2).

Mean disease severity progress curves for both soils were adequately described by the logistic model (pseudo-*R*^2^ ≥ 0.98) (Figure 2). 

The estimates for the upper asymptote for each of the soils (clay and sandy loam) were respectively: 44.9 and 21.0 for unfertilized-daily; 45.3 and 17.1 for unfertilized-deficit; 69.8 and 27.3 for NPK-daily; 68.1 and 40.3 for NPK-deficit; 90.6 and 62.0 for N-daily; and 68.1 and 60.9 for N- deficit treatments. Similarly, estimates for the rate of MS increase for each of the soils were respectively: 0.07 and 0.09 for unfertilized-daily; 0.05 and 0.07 for unfertilized-deficit; 0.08 and 0.12 for NPK-daily; 0.07 and 0.05 for NPK-deficit; 0.11 and 0.07 for N-daily; and 0.05 and 0.04 for N-deficit treatments. Highest estimates parameters were found in the daily irrigated plants, irrespectively of the fertilization treatment. Thereafter, high asymptote parameters were fitted to plants N-daily treated in the two soils, as well to NPK-daily treatment in the clay soil. The estimate rate parameters were also high in the clay soil of N-daily treatment, and in the NPK-daily treatment in sandy loam soil (Figure 2). The final value of severity (MS parameter) in clay soil was 54.5 % significantly higher in plants N-fertilized than the mean value obtained in plants unfertilized and fertilized with NPK (Table 1). In sandy-loam soil, MS values were 53.6% higher in plants fertilized (79.2% with N and 73.1% with NPK, respectively) than plants unfertilized (35.3%) (Table 2).

## 3. Discussion

In general, plant nutrition affects growth and its defense mechanisms against pathogen attack [31]. More specifically, regarding Verticillium wilt, the use of low doses of N, as well as the reduction of Ca levels in the soil, or the increases of K and Mg seem to be associated with lower incidences and severities of the disease [32]. These results mostly come from studies in herbaceous plants [8,9,11,15]. However, similar studies have been scarce in woody crops affected by Verticillium wilts [17,18,19], particularly the olive tree. In relation to this woody crop, recent artificially performed experiments by López-Moral et al. (2022) [20] showed that fertigation with N excess once per week significantly reduce the relative area under disease progress curve compared with control (olive cuttings watered with water). However, in line with the results found in this study, many Andalusian olive technicians and farmers have perceived that certain imbalance in fertilization, particularly N-K, or excessive applications of N, are favoring infections by *V. dahliae* (L. F. Roca Castillo; personal communication). 

In this study, the influence of the fertilization formulations NO_3_Ca (15.5%-0-0) hydrosoluble (N) and N-P-K (15%-15%-15%) hydrosoluble (NPK) and its interaction with the frequency of irrigation (two irrigation frequencies: daily and deficit) on the development of Verticillium wilt has been evaluated in susceptible Picual olive cultivar in microplots with two naturally infested soils of different texture. Under the aforementioned conditions, ‘Picual’, chosen for being one of the most important economically (and historically) worldwide [1], showed differential reactions depending on the presence or not of *V. dahliae*. In sterile microplots, no differences in growth were observed in plants subjected to different fertigation treatments when they were grown under the aforementioned conditions. The opposite pattern was reported in control pots of a similar Olive/*V. dahliae*/irrigation study under natural environmental conditions with Picual cultivar [33], but also in a Olive/irrigation study under field conditions with ‘Cornicabra’, where growth parameters were significantly reduced with deficit irrigation treatments [34]. It is hypothesized that at daily treatment, the amount of water supplied in this trial was perhaps above that required, in terms of growth, for a woody plant that is adapted to the dry Mediterranean climate. Thus, deficit applied here is closer than daily to the amount required for maximum growth for olive trees. However, in infested microplots, ‘Picual’ showed differential reactions depending on the type of soil. 

In clay soil, the mean value of RAUDPC calculated in plants fertilized with N were significantly higher (70.0%) than plants unfertilized and plants fertilized with NPK independently of frequency of irrigation. Under the same environmental conditions (including soil), Pérez-Rodríguez et al. (2015a) [26] found significant differences for the final values of RAUDPC between irrigation treatments in the susceptible Picual olive, with the daily irrigation schedule being significantly greater than the rest of the treatments (weekly, biweekly and deficit). This could indicate that the fertilization factor had a greater influence on VWO than irrigation under the environmental conditions used in this study. Likewise, it could also indicate that the use of different Nitrogen sources can influence the degree of susceptibility of plants to wilting. In this sense, some authors argue that the radical NH_4_-N may cause a decrease in the number of microsclerotia, the infective, dispersal and survival structures of *V. dahliae* 9]. This decrease may be due to the toxins generated by this radical on the pathogen [35], or by the stimulus it exerts on this element in the growth of antagonist flora [36]. Conversely, the radical NO_3_-N may increase the amount of MS according to Muromtsev and Chernyaeva (1979) [37] and Chernyayeva et al. (1984) [36]. This could explain the higher RAUDPC values found in plants N-fertilized compared with plants NPK-fertilized, as the formulation of the former contain higher amount of this radical (N and NPK treatments contain 14.5% and 7% of NO_3_-N, respectively).

However, in the sandy loam soil there was a significant fertilization-irrigation interaction and Picual trees N-daily treated reached final RAUDPC values that were significantly higher than the rest of treatments. It seems that the influence of the mentioned factors was equitable when the physicochemical characteristics of the soil were different from those of a clay soil. In this case, the influence of these factors could be indirect affecting the inoculum density of *V. dahliae* in the soil. Multivariate analyzes carried out on more than 80 soil samples from the Iberian Peninsula related a high level of inoculum density of *V. dahliae* with a relative percentage of clay greater than 31.0% [38]. Likewise, López-Escudero et al. (2010) [39] reported a significantly greater incidence of the disease in olive orchards established on Vertisol soils, or soils with a clay content greater than 30% [40], than that registered in plantations established in soils with a lower clay content (Alfisol). However, the final mortality in plants treated in this study with daily N was 37% and 85.2% in clay and sandy loam soils, respectively. Other causes relating to the direct or indirect effect of fertigation on the plant could influence the radical infection and/or vascular colonization; these aspects need to be elucidated by further research. 

On the other hand, soil texture did not affect the parameter final disease incidence, being 100% in plants grown in clay and sandy loam soils and N-daily treated. For this disease parameter, it seems that the effect of both factors (fertilization and irrigation) was additive. In this sense, the incidence of Verticillium wilt in other woody hosts (pistachio) has been significantly associated with deficiencies of potassium in soils with low inoculums densities of *V. dahliae* [18]. Available K in both soils was the same (415 ppm), but NPK treatment contain 15% of potassium Oxide (K_2_O) while N treatment did not contain this element. Complex interactions could have caused the disease incidence parameter to be so high and equal between soils.

For efficient integrated management of VWO, it is important to clarify the individual influence of fertilization and irrigation, and interactions with each other, since they are one of the main culture practices that currently affect this disease in modern olive orchards. This research provides novel information about the influence of nutritional supply via irrigation (fertigation) on the occurrence and development of VWO under semicontrolled conditions, using microplots that were naturally infested by *V. dahliae*. One of the major conclusions of this work is that the application of nitrogenous fertilizers or the application of nitrogenous fertilizers with daily frequency irrigations in naturally infested soils strongly encourages the onset and development of VWO in susceptible Picual olive (Table 1 and Table 2; Figure 1 and Figure 2).

## 4. Materials and Methods

### 4.1. Experimental Plot

Studies were carried out under a semicontrolled environment in an experimental plot sited in Campus de Rabanales (Universidad de Córdoba, Spain) (37°54′58.2″ N 4°42′53.1″ W) (See Appendix A for details). A row of 40 cement containers with a capacity of approximately 1 m^3^, covered by a galvanized sheet roof to protect them from rain and excessive exposure to the sun, and with a North-South orientation was built on the ground in this plot. These microplots were open at the bottom to prevent the accumulation of water at the bottom and facilitate the growth of plant roots.

This plot consisted of a line of 40 concrete containers located above ground. Each plot was 1 m^2^ by 70 cm deep, open at the bottom, North-South oriented, and protected from rain and excessive sun by a galvanized steel cover. The microplots were filled with two different soils naturally infested with *V. dahliae* that were planted with the susceptible olive cultivar ‘Picual’. During experiment plants were fertilized with different doses of nitrogen compounds and irrigated with different irrigation frequencies as explained below.

### 4.2. The Soils: General Characteristics, Inoculum Density of the Pathogen and Physical and Chemical Properties

#### 4.2.1. General Characteristics

Both soils were collected in Andalucía, southern Spain (Appendix A). The first naturally infested soil (Soil 1) come from a field located in the municipality of Utrera (lower Guadalquivir Valley, Sevilla province). This is a marsh area annually cropped with *V. dahliae* hosts such as cotton, sugar beet, tomato, eggplants, and other vegetable crops. The second one (Soil 2) came from a field located in the term of Villanueva de la Reina (high Guadalquivir Valley, Jaen province) that had been continuously cultivated with cotton during several previous years. In the two zones where soils were collected VWO incidence and mortality develops quickly, as demonstrated by Trapero et al. (2013) [41] for Soil 1 or by the results of experiments within the frame of the Olive Breeding Program of the University of Córdoba [42] for Soil 2.

#### 4.2.2. Inoculum Density of the Pathogen

Population of *V. dahliae* in soils was assessed by the wet sieving technique [43] using as culture media a modified sodium polypectate agar medium [44]. Ten replications (petri plates) for analyses were used. Results were expressed as propagules or microsclerotia, the infective, dispersal and survival structures of the pathogen, per gram of air-dried soil. After analysis, inoculums density of the pathogen accounted for 20 and 9 microsclerotia/g for Soil 1 and 2, respectively. The isolation and molecular characterization of the pathogen conducted according the methodology described by López-Escudero and Blanco-López (2005) [45] and Mercado-Blanco et al. (2003) [46], revealed that highly virulent strains of *V. dahliae* (defoliating pathotype) were prevalent in both soils.

#### 4.2.3. Physical and Chemical Properties

Soil 1 was a clay soil (pH in ClK = 7.47, organic matter = 1.94%, total CaCO_3_ equivalent = 13.75%, available K = 415 ppm, and available P = 55.4 ppm), with a bulk density of 1.300 Kg/m^3^. Soil 2 was a sandy-loam soil (pH in ClK = 7.43, organic matter = 1.04%, total CaCO_3_ equivalent = 19.69%, available K = 415 ppm, and available P = 55.4 ppm), with a bulk density of 1.500 Kg/m^3^. 

#### 4.2.4. Hydraulic Properties

To characterize the hydraulic properties of the soils, a 12-L plastic pot filled with soil was watered until saturation was reached, then allowing the excess water to percolate freely through the bottom. After weighing the pots, they were dried for one week at 70 °C in an oven. The pot was then weighed again to calculate the mass of water and the mass of solid soil. The gravimetric method was used to calculate the upper limit of soil water content (θ_UL_) according to the formula: θ_UL_ = (Vw)/(Vs), where Vw = mass of water and Vs = total volume of soil. The calculated θ_UL_ values for Soils 1 and 2 accounted for 0.37 and 0.31 m^3^ m^−3^, respectively. Moreover, the values of lower limit of the soil water content (θ_LL_) were calculated for two soils with ye aim to know the available water in soil. This was made by sowing barley seeds in a 12-L plastic pot filled with soil, where plants grew until reaching a maximum vegetative development. Then, the pots stopped watering, plants died, and the soil water content was calculated according to the above formula. The θ_LL_ values for Soils 1 and 2 reached 0.17 and 0.11 m^3^m^−3^, respectively.

### 4.3. The Plant Material

For the experiments, one-year-old rooted olive cuttings of cultivar Picual were used (Appendix A). This cultivar is the most important olive cultivar worldwide, it is susceptible to the disease, and it is routinely used as the type cultivar in most of the VWO epidemiology and control experiments carried out by our group at artificially inoculations and at field and semi-controlled conditions [1,26,27,42,47,48]. Plants came from a commercial nursery that produce rooted olive cutting from certificated mother olive plants free of *V. dahliae* and other olive pathogens. At planting time, plants were 0.6–0.8 m high, with a single trunk and three or four secondary branches. 

### 4.4. Microplot Establishment, Experimental Design, and Treatments

In January 2015, approximately 36 tons per soil form the two collection areas (Utrera Marsh and Villanueva de la Reina) were transported to the experimental site in Rabanales. Soils were separately homogenized by continuous crumbling and mixing using a tractor shovel. For each soil, 20 microplots were filled with 800 kg of soil. Two of the microplots of each soil were filled with sterile soil, remaining as disease controls. For this, 1.6 tons per soil were sterilized by autoclaving twice at 120 °C for 70 min (S1000 steam sterilizer; Antonio Matachana S.A., Barcelona, Spain). In February 2015, nine olive plants were transplanted per microplot (Appendix A–F).

A split-split-plot design (2 × 3 × 2 factorial combination of treatments) was used to study the influence of irrigation frequencies, nitrogen fertilization, and soil (the two ones described above) on the onset and development of VWO. The main plot was the irrigation treatment, the sub-factor was the fertilization, and the sub-sub-factor was the soil. 

The two irrigation frequencies were daily and deficit irrigation. The irrigation schedule for these two treatments was established according to the ‘Relative Soil Water Deficit’ (RSWD) values, calculated by applying the formula: RSWD = (θ_UL_ − θi)/(θ_UL_ − θ_LL_), where θi is the volumetric soil water content [26]. For both treatments the RSWD parameter was generally less than 0.9. 

For its part, the three nitrogen fertilization treatments evaluated were: (1) Control without fertilization (Unfertilized); (2) Fertilization with N-P-K (15%-15%-15%) hydrosoluble (Total nitrogen = 15%, Nitrate = 7%, Ammonium Nitrogen = 8%, Phosphorus Pentaoxide (P_2_O_5_) = 15%, Potassium Oxide (K_2_O) = 15%, added with micronutrients; Dose 1 g/l) (NPK); and (3) Fertilization with Calcium Nitrate (15.5%-0-0) hydrosoluble (Total nitrogen = 15.5 %, Nitrate Nitrogen = 14.5%, Ammoniacal Nitrogen = 1% and 19% Soluble Calcium as Ca; Dose 2 g/L) (N). Non-infested microplots subjected to the daily frequency irrigation and fertilization treatment NPK were used as control. 

For each microplot, the watering system consisted of a stopcock, six branch lines (16 mm in diameter, 80 cm in length, and with 16.5 cm spacing), and thirty-six compensating drippers (2 L/h, 6 per branch) (Netafim TM). Water for unfertilized microplots was dispensed through a PVC water pipe (50 mm diameter and 44 m long) extended along the container line that was connected to a programmable irrigation controller (Image 6 Rain Bird, Rain Bird Iberica S.A., Spain). On the other hand, water plus fertilizer for fertilized microplots was dispensed through another PVC water pipe parallel to the previous and also extended along the microplot line by a water pump (Ferca Pumps, AP 1500M 1.5HP, Boqueixón, A Coruña, Spain), with a maximum flow of 80 L/min, connected to 2 cylindrical mixing tanks of 1000 L capacity (Appendix A).

### 4.5. Soil Water Content Measurements in the Microplots

The soil water content in each microplot was recorded every fifteen days using a Time Domain Reflectometry (TDR) system (6050X1 Trase System I; Soil Moisture Equipment Corp., Santa Barbara, CA, USA). TDR probes were stainless steel welding (6 mm in diameter) cut as pairs to 30 cm in length. Two pairs of rods were inserted vertically in each microplot at a parallel distance of 50 mm. This procedure resulted in a single water-content reading integrated over depth traversed by the probes. Data were used for planning an irrigation schedule that was individually calculated for each treatment and microplot. The volumetric water content of each container (RSWD level) was maintained corresponding to its irrigation treatment. 

### 4.6. Disease Progression and Tree Development

The experimental plot was surveyed periodically for 22 months after planting for disease symptoms, particularly during the most favorable environmental periods for disease development (for our conditions, these periods were spring, early summer and fall). Disease severity was scored weekly based on the percentage of affected plant tissues such as leaves and shoots showing symptoms of chlorosis, necrosis and/or defoliation. To this end, plants were assessed weekly for 14 weeks after inoculation using a 0 to 16 rating scale. This scale estimated the percentage of affected tissue using four main categories or quarters (≤25, 26–50, 51–75, and 76–100%) with four values per category. Thus, each scale value represents the number of sixteenths of affected plant area. The scale values (X) are linearly related to the percentage of affected tissue (Y) by the equation Y = 6.25X − 3.125 [30]. These values were used to build the progress curves for the disease incidence (DI) of the affected plants and the mean severity (MS) of the symptoms over time during the recording period. For analyzing the temporal progress of the disease, curves of MS over time were adjusted to the logistic model
(1)$$y_{L}\left(t\right)=\frac{\rho_{1}}{\left\{1+exp\left[-\left(\rho_{2}+\rho_{3}t\right)\right]\right\}}$$
where *ρ*_1_ is the upper asymptote, *ρ*_2_ is related to initial level of disease severity, and *ρ*_3_ is rate of the process [49,50]. 

At the end of the experiment, the area under the disease progress curve (AUDPCP) was estimated as the percentage of the maximum possible value in the considered period according to the following formula based on Campbell and Madden (1990) [51]: AUDPCP = [(*t*/2 × (*S*_2_ + 2 × *S*_3_ + … + 2 × *Si-_1_* + *Si*)/4 × *n*] × 100, where *t* = the interval in days between observations; *Si* = the final mean severity; 4 = the maximum disease rating; *n* = the number of observations. 

The final percentage of dead plants, or mortality (M), was also considered in estimating the severity of the reactions. The time that elapsed from transplanting to the onset of disease symptoms (time establishment, FS) and the mean mortality time (MM) (mean time between the onset of disease symptoms and plant death) were determined for all irrigation/fertilization combinations, as described by Pérez-Rodríguez et al. (2015a) [26]. 

To avoid the incorporation of new inoculum into the soil, falling leaves and dried plant debris from dead shoots and buds from affected plants in infested microplots due to slow decline or apoplexy syndromes caused by pathogen infection [1] were removed periodically from the surface of the microplots.

### 4.7. Pathogen Isolation

Plant infection was confirmed by isolating the fungus from the affected shoots or leaf petioles of diseased plants. Affected woody tissue samples were washed in running tap water, and the bark was removed. The tissue was surface was then disinfected in 0.5% sodium hypochlorite for 1 min. Wood chips were placed on PDA plates and incubated at 24 °C in the dark for 6 days. When affected tissues did not yield detectable pathogen cultures, shoot tissues were analyzed using PCR methods as described by Mercado-Blanco et al. (2003) [46].

### 4.8. Final Inoculum Density of the Pathogen in Microplots

The inoculum density of the pathogen in the soil was also assessed at the end of the experiment for all individual microplots and for all treatments. A 100 g soil sample was taken from the 0 to 25 cm depth profile from each container using a cylindrical auger. The ID was quantified using the method described above. 

### 4.9. Statistical Analysis

In this trial, an analysis of variance (ANOVA) of final values of the disease parameters DI, MS, M and AUDPC was performed. The mean values of analyzed parameters were compared using Fisher’s protected least significant difference test at *p* = 0.05.

By its side, the TE and MM parameters were analyzed using Kaplan-Meier survival analysis [52], in which the survival times were calculated as the week in which a plant died or showed disease symptoms for the first time. Comparisons were tested using the log-rank test at *p* = 0.05. 

The Akaike’s information criterion and pseudo*-R*^2^ parameter were used to evaluate the appropriateness of the logistic model to describe data. 

Once the experiment was concluded, the combined analysis (Split-split-plot) of variance for values of AUDPC and MS revealed that the four analyzed factors (irrigation, fertilizer, soil-fertilizer interaction, and soil) were significant, with *p* values < 0.05. Significance of the interaction was due to use two different soils. For that, each soil was studied separately by new ANOVAs and Fisher’s test used for mean comparisons.

Statistix 10.0 software program (Analytical Software, Tallahasse, FL, USA) was used for all mentioned analyses.

## Figures and Tables

**Figure 1 plants-11-03551-f001:**
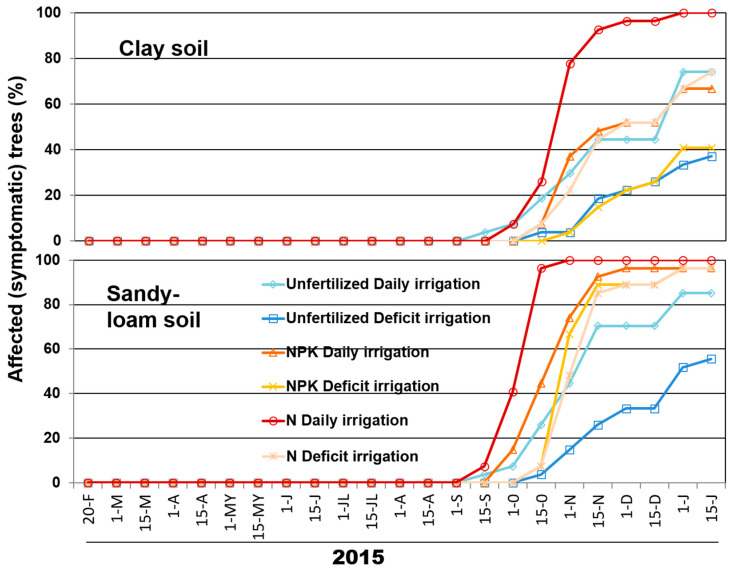
Verticillium wilt disease incidence over time in Picual cultivar olive plants grown in microplots of two naturally infested soils (for more details see Materials and methods section) with *Verticillium dahliae* and subjected to a combination of two irrigation frequencies: daily and deficit, and three nitrogen fertilization treatments: control without fertilization (Unfertilized), fertilization with N-P-K (15%-15%-15%) (NPK) and fertilization with Calcium Nitrate (15.5%-0-0) (N).

**Figure 2 plants-11-03551-f002:**
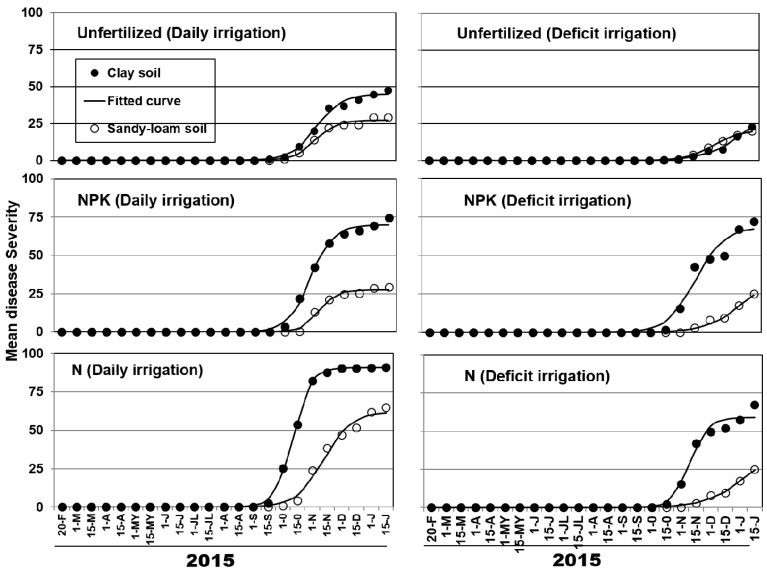
Progress of mean severity of symptoms of Verticillium wilt in Picual cultivar olive plants grown in microplots of a clay and sandy-loam soils (for more details see Materials and methods section) naturally infested with *Verticillium dahliae* and subjected to a combination of two irrigation frequencies: daily and deficit, and three nitrogen fertilization treatments: control without fertilization (Unfertilized), fertilization with N-P-K (15%-15%-15%) (NPK) and fertilization with Calcium Nitrate (15.5%-0-0) (N). Disease severity was estimated on a 0 to 16 rating scale. This scale estimated the percentage of affected tissue using four main categories or quarters (≤25, 26–50, 51–75, and 76–100%) with four values per category. Thus, each scale value represents the number of sixteenths of affected plant area. The scale values (X) are linearly related to the percentage of affected tissue (Y) by the equation Y = 6.25X − 3.125 [30]. Symbols correspond to the observed mean severity data and lines are the estimated logistic curves.

**Table 1 plants-11-03551-t001:** Effect of fertilization treatments and irrigation frequency on the final estimated disease values for susceptible cultivar Picual in microplots of clay soil naturally infested by *Verticillium dahliae*
^a^.

Fertilization	Irrigation	FS (Range) ^b^	Mortality (%) ^c^	MS (%) ^d^	Incidence (%)	MM (Weeks) ^e^	RAUDPC ^f^
Unfertilized	Daily	40.6 a (31–48)	18.5 a	29.1 a	74.1 a	3.7 a	51.3 a
	Deficit	42.9 a (35–50)	7.4 a	20.7 a	37.0 a	6.0 a	22.9 a
	Mean	41.7 A	12.9 B	24.9 B	55.5 A	4.8 A	37.1 B
NPK	Daily	39.3 a (35–46)	18.5 a	29.3 a	66.7 a	6.0 a	17.5 a
	Deficit	42.7 a (37–46)	3.7 a	24.9 a	40.7 a	6.0 a	20.7 a
	Mean	41.0 A	11.1 B	27.1 B	53.7 A	6.0 A	34.1 B
N	Daily	41.5 a (35–50)	37.0 a	64.7 a	100.0 a	7.0 a	89.2 a
	Deficit	37.3 a (33–46)	25.9 a	49.4 a	74.1 a	7.7 a	50.8 a
	Mean	39.4 A	31.4 A	57.1 A	87.0 A	7.3 A	70.0 A

^a ^Trees were surveyed periodically for symptom development over 11 months (February 2015 to January 2016). The values listed in columns followed by the same letter were not significantly different at *p* = 0.05. ^b^ FS = The time (weeks) from orchard establishment to the onset of disease symptoms. Differences were estimated according to the log rank test. ^c^ Mortality (%) = Differences were estimated according to Fisher’s protected (LSD) test. ^d^ MS = Mean severity in percentage. Differences were estimated according to Fisher’s protected (LSD) test. ^e^ MM = Time from symptom onset to plant death. Differences were estimated according to the log rank test. ^f^ RAUDPC = Relative area under disease progress curve expressed as a percentage of maximum theorical curve.

**Table 2 plants-11-03551-t002:** Effect of fertilization treatments and irrigation frequency on the final estimated disease values for susceptible cultivar Picual in microplots of sandy-loam soil naturally infested by *Verticillium dahliae*
^a^.

Fertilization	Irrigation	FS (Range) ^b^	Mortality (%) ^c^	MS (%) ^d^	Incidence (%)	MM (Weeks) ^e^	RAUDPC ^f^
Unfertilized	Daily	38.6 a (31–48)	25.9 a	47.6 a	85.2 a	5.8 a	37.8 c
	Deficit	41.8 a (35–50)	0.0 a	23.0 a	55.6 b	…	9.6 d
	Mean	40.2 B	12.9 B	35.3 B	70.4 A		23.8 B
NPK	Daily	36.4 a (33–42)	48.1 a	74.4 a	96.3 a	5.7 a	61.1 b
	Deficit	38.0 a (35–46)	48.1 a	71.9 a	96.3 a	9.1 a	44.9 c
	Mean	37.2 A	48.1 A	73.1 A	96.3 A		52.9 A
N	Daily	34.1 a (31–46)	85.2 a	90.9 a	100 a	3.7 b	94.3 a
	Deficit	38.8 a (35–46)	22.2 a	67.5 a	96.3 a	6.8 a	42.6 c
	Mean	36.4 A	53.7 A	79.2 A	98.1 A		68.4 A

^a ^Trees were surveyed periodically for symptom development over 11 months (February 2015 to January 2016). The values listed in columns followed by the same letter were not significantly different at *p* = 0.05. ^b^ FS = The time from orchard establishment to the onset of disease symptoms. Differences were estimated according to the log rank test. ^c^ Mortality (%) = Differences were estimated according to Fisher’s protected (LSD) test. ^d^ MS = Mean severity in percentage. Differences were estimated according to Fisher’s protected (LSD) test. ^e^ MM = Time from symptom onset to plant death. Differences were estimated according to the log rank test. ^f^ RAUDPC = Relative area under disease progress curve expressed as a percentage of maximum theorical curve.

## Data Availability

All data analyzed in this study are included in this article.

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
