# Peer review of "High Input of Nitrogen Fertilization and Short Irrigation Frequencies Forcefully Promote the Development of Verticillium Wilt of Olive"

_plants, 2022, doi:10.3390/plants11243551_

Round 1

Reviewer 1 Report

In this work, the authors describe that a high supply of nitrogen and high irrigation favor Verticillium wilt of olive trees in potted plants in infested soils. This is not surprising because the pathogen grows and spreads in the vascular system of the plant, and depends on the adequacy of water, carbon and nitrogen compounds. Therefore, the growth of Verticillium dahliae is favored by any environmental conditions that contribute to the growth of the plant. This is easy to see in figures 1, 2 and 3. The plants were potted in spring to be infected. During the summer, the high temperatures stop the growth of the plants and pathogen, so there are no symptoms of disease. Once autumn arrives with mild temperatures, the pathogen grows and the symptoms of Verticillium Wilt were detected. The diseases progress until the cold comes when winter approaches and then the metabolism of the plant stagnates. Therefore, I find a low novelty in this work.

Other points:

1.       Line 61 same reference twice.

2.       Line 65, first time MS abbreviation, please define.

3.       Table 1, please use the same format than table 2. Specially take care about second column.

4.       Line 157, first time DI abbreviation, please define.

5.       Figure 1, I suggest change Soil1 by Clay-soil and Soil 2 by Sandy-loan soil. It will be easy to understand.

6.       Please combine Figures 2 and 3 by representing together for each soil, Daily irrigation and Deficit irrigation, using solid and open symbols. It is better to see the results in both types of soils in the same figure.

Reviewer 2 Report

The authors present an interesting manuscript entitled: "High inputs of nitrogen fertilization and short irrigation frequencies forcefully promote the development of Verticillium Wilt of olive". This research work is very interesting in its subject matter and also has a direct application in the field, not only in olive cultivation, but also in other similar crops. The economic losses produced by this disease are high and the search for strategies that can minimize its incidence are of great interest.

The manuscript and study are well written, easy to follow in almost all parts, and scientifically well structured. The main problem with this type of studies is that they are highly dependent on development conditions. There are many nutritional parameters that are combined and many times it is not only the effect of one of them, but the whole that leads to interesting results regarding the control of the incidence of this disease.

In my opinion the authors have presented conclusions that are valid and relevant based on the results obtained. I consider that the manuscript can be accepted for publication in the journal after answering and making some minor recommendations that are detailed below:

1.- Introduction section is well conducted and it show the state or art correctly. No changes to the content are recommended.

-          Line 11: Verticillium must be in capital.

-          Line 15: NO3Ca modified by NO3Ca. Change it along the manuscript.

2.- Results.

-          Line 97: authors write the abbreviature ID but it is not explained until line 446. Please include the means the first time. Besides, correct way is DI as explain at line 446.

I would recommend a restructuration of the section in this part of the manuscript. In my opinion, subsection 2.2 and 2.3 could be merged. At the same time, subsection 2.1 could be include along the results detailed in current subsection 2.4.

In relation to Table 1 and Table 2, I would recommend a revision of the format so that all the data could be entered in a single line, expanding some columns. At the same time, I understood that Unfertilized (daily) means: no treatment and irrigation daily, but I do not understand why is Daily or Deficit in parenthesis? I consider a new redaction of this column. The same in Table 2.

By other hand, I understood the table after several reading because it is not clear along columns and files that MEAN is related to the two previous values.

In general, although the table could be understood, I would recommend a new format or presentation of the results in both tables.

I would like also to see some pictures along the manuscript or even an appendix with pictures of olive plots, disease progression, culture of Verticillium in vitro, symptoms, etc… I recommend to include some figures along Results and Material and Methods.

-          Paragrah between lines 119 – 122. Clarify the information of this paragraph. At the current text, it seems that in both cases the symptoms appeared at 31 weeks, but the authors say However... Clarify this information.

In relation with Figure 1, it would be clear using colours. In the case of soil 1, the line corresponding to Unfertilized Daily irrigation is not with the filled square but with the transparent box, so it is not as the legends.

3.- Discussion

This section is well conducted and the conclusion are clear but I would like to ask authors above the lack of N and water. Are there studies in which we can see the effects on the normal development of the plants, as well as on production levels.

A priori it is easy to imagine that with less water, therefore with less humidity, the appearance of symptoms of the disease is delayed, since the fungus also needs humidity. But are there studies where the effects on the plant and the production of this lack of irrigation are seen?

In my opinion could be very interesting to discuss about the repercussion along the development of plants and production levels of the treatment recommended.

-          Line 293-294: López Escudero et al. 2010 is repeated.

4.- Material and Methods

This section is correct in my opinion. Subsection 4.2.4. contains some results, but it is difficult to include a subsection in results only for these two data.

Once these suggestions are taken into account by the authors, I would like to recommend the manuscript for publication with minor revision.

Round 2

Reviewer 1 Report

Minor issues:

Figure 1 legend. Please remove the Soil 1 and Soil 2 mentions. They have no sense know.

Same coment for Figure 2 legend.

Author Response

Point 1. Figure 1 legend. Please remove the Soil 1 and Soil 2 mentions. They have no sense know. Same coment for Figure 2 legend.

Response 1. Thank you for the correction. Soil 1 and Soil 2 mentions were deleted from both Figure 1 and 2 legends.